# OpenReview forum: "HEARTS: A Holistic Framework for Explainable, Sustainable and Robust Text Stereotype Detection"
_NeurIPS.cc/2024/Workshop/SafeGenAi — SafeGenAi Poster_

### Official Review · Reviewer_Bjgy · 2024-10-08
**An Interesting Framework for Explainable Stereotype Detection**

**Rating:** 7
**Confidence:** 4

**Review:**

This paper introduces HEARTS, a framework for detecting stereotypes in text that aims to improve explainability, sustainability, and robustness. The authors present a new dataset, EMGSD, and use it to fine-tune BERT models for stereotype detection. They then analyze their best-performing model using explainability techniques and apply it to assess bias in large language model outputs.

The work addresses an important problem in natural language processing and AI ethics. Stereotype detection is challenging, especially given the subjective nature of stereotypes across different cultures and contexts. The authors' approach of combining improved datasets, efficient models, and explainability techniques is sensible and shows promise.

Pros:
(1) The EMGSD dataset is a valuable contribution. It expands on existing datasets by including underrepresented groups like LGBTQ+ and regional stereotypes. This should help improve the generalizability of models trained on it.

(2) The focus on model efficiency and carbon footprint is commendable. The authors demonstrate that their ALBERT-V2 model achieves good performance with significantly lower emissions than larger BERT variants.

(3) The explainability framework, combining SHAP and LIME, provides useful insights into the model's decision-making process. The example analyses in the paper help illustrate how this can be applied in practice.

Cons:
(1) The paper could benefit from a more thorough discussion of the limitations of the EMGSD dataset. For example, the authors mention that racial minorities are still underrepresented, comprising only about 1% of the sample. It would be helpful to see a more detailed breakdown of the dataset composition and a discussion of potential biases this might introduce.

(2) The process of creating neutral and unrelated sentences for the dataset using GPT-4 raises some concerns. While the authors state they performed manual review, more details on this process and its potential limitations would strengthen the paper. For instance, how many samples were manually reviewed? Were there any patterns in GPT-4 generated content that required adjustment?

(3) The evaluation of LLM bias seems somewhat limited. While the authors test 12 models, they only use 35 prompts (5 per group and dataset combination). This seems like a relatively small sample size given the complexity and variability of LLM outputs. A larger set of prompts or a discussion of how representative these prompts are would be beneficial.

(4) The paper would benefit from more discussion of potential real-world applications and limitations of the HEARTS framework. For example, how might this be integrated into content moderation systems? What are the potential risks of false positives or negatives in such applications?

(5) The comparison between SHAP and LIME explanations could be expanded. While the authors provide some examples and aggregate statistics, a more in-depth analysis of when and why these methods disagree could provide valuable insights.

---

### Official Review · Reviewer_B5wv · 2024-10-09
**HEARTS: A Holistic Framework for Explainable, Sustainable, and Robust Text Stereotype Detection**

**Rating:** 8
**Confidence:** 3

**Review:**

## Summary
The paper introduces HEARTS, a comprehensive framework for detecting and explaining stereotypes in text using machine learning models. The framework also emphasizes sustainability and robustness, using methods like SHAP and LIME for explainability, and ALBERT-V2 for carbon-efficient fine-tuning. The paper presents the Expanded Multi-Grain Stereotype Dataset (EMGSD), a significant enhancement of previous stereotype datasets, and shows that HEARTS outperforms current approaches across multiple benchmarks.

## Strengths
* Novelty: The HEARTS framework is a comprehensive and well-rounded approach to stereotype detection, combining explainability, robustness, and sustainability in a unique manner. The addition of carbon-efficient training processes is a strong point that aligns with modern concerns about the environmental impact of machine learning.

* Expanded Dataset: The introduction of the EMGSD, which incorporates previously under-represented demographics such as LGBTQ+ and certain racial and nationality stereotypes, is a significant contribution. The dataset is well-constructed, showing a strong methodology for filtering and augmenting data from the WinoQueer and SeeGULL datasets.

* Explainability: The use of SHAP and LIME to explain the predictions made by the models at the token level is innovative. By generating explainability confidence scores and ensuring alignment between SHAP and LIME outputs, the framework enhances trust and interpretability in stereotype detection.

* Sustainability: The choice to use ALBERT-V2 for stereotype classification reduces the model’s carbon footprint. The study provides detailed carbon emission comparisons, making this paper relevant to researchers focusing on environmentally sustainable AI.

* Comprehensive Evaluation: The paper evaluates stereotype prevalence in outputs from 12 different LLMs using a robust experimental setup. This provides valuable insights into the biases of major models, including LLaMA and GPT-4 series.

## Areas of improvement
* Generalization Across Tasks: While the HEARTS framework performs well on stereotype detection, it remains unclear how well it can generalize to other domains, especially tasks unrelated to stereotype detection. The reliance on specialized datasets like EMGSD limits the scope of application.

* Dataset Bias: Despite efforts to improve demographic diversity, the paper acknowledges that the EMGSD dataset still under-represents certain groups, particularly racial minorities, which could skew the performance of the models on those groups.

* Explainability Methods: The paper heavily focuses on SHAP and LIME for explainability, but these methods have known limitations, especially in terms of scalability to larger models. The reliance on these methods may restrict the framework’s utility in real-time or large-scale applications.

* Model Performance Variability: The paper shows that the ALBERT-V2 model’s performance varies significantly across different demographic groups, with weaker results for detecting gender and profession stereotypes compared to LGBTQ+ stereotypes. This suggests that more work is needed to improve model generalization across diverse stereotype types.

* Complexity of Methods: The proposed framework, while innovative, is somewhat complex in its reliance on multiple tools for explanation (SHAP, LIME) and the dataset augmentation process. Simplifying these methodologies or providing clearer steps could improve the framework's practicality for broader use.

## Conclusion
The HEARTS framework is a well-thought-out and impactful contribution to the fields of stereotype detection and explainable AI. Its focus on under-represented groups, carbon-efficient training, and explainability makes it a timely and valuable addition to the existing literature. While there are some areas for improvement, particularly in generalization and scalability, the paper provides a solid foundation for future work in both stereotype detection and responsible AI

---

### Official Review · Reviewer_q5oB · 2024-10-09
**Novel analysis of token-level detection of (harmful) stereotypes using similarity between XAI metrics**

**Rating:** 7
**Confidence:** 4

**Review:**

The authors execute of token-level detection of (harmful) stereotypes, offering a novel approach for detection and applying the task to a novel collection of datasets. Their approach uses fine-tuned ALBERT-v2 model to generate classifications and token-level scores using SHAP and LIME. They use the similarity of the SHAP and LIME score vectors to generate a confidence score.

Overall the work is interesting and motivated by real problems facing the field; stereotype detection is challenging, we have an overreliance on large models where fine-tuned smaller ones will do, and we need to think more about what confidence means.

The main point where I want to challenge the authors is to think more about the "confidence" that they are calculating.

Typically confidence scores are derived from statistical uncertainty. For the author's method, I think what they are identifying is uncertainty due differences in XAI metrics. There is some intuitive sense to this; in theory, something that is "certain" should be scored more similarly across different "good" XAI metrics. But how should we interpret a low score in this context? It could be:

a. The labeled text represents something that is genuinely ambiguous, meaning that the XAI metrics are going to have high variance.
b. One XAI metric fails to correctly extract the "explanation" from the model for this text
c. Both metrics fail

If I were working on this question, I would want to see how this approach corresponds with other, less ambiguous sources of uncertainty such as:

d. Recalculating the class proportions over different bootstrapped samples of the test split
e. Training models from different starting seeds

These get at different but related sources of uncertainty that the authors should compare.

Addendum:

- What is `llama-3.1-405b-T`? (Namely, what is the `-T` suffix?)